# Preparation of Hierarchical Porous Carbon Aerogels by Microwave Assisted Sol-Gel Process for Supercapacitors

**DOI:** 10.3390/polym11030429

**Published:** 2019-03-06

**Authors:** Xueqing Cai, Guiming Tan, Zhentao Deng, Jianhong Liu, Dayong Gui

**Affiliations:** College of Chemistry and Environmental Engineering, Shenzhen University, Shenzhen 518055, China; xueqing_cai1026@163.com (X.C.); tanguiming530@126.com (G.T.); 18316787927@163.com (Z.D.); liujh@szu.edu.cn (J.L.)

**Keywords:** microwave, carbon aerogels, hierarchical porous, KOH activation, supercapacitor

## Abstract

Low-cost resorcinol formaldehyde (RF) organic aerogels were prepared by using resorcinol and formaldehyde as precursors, and sodium hydroxide as a catalyst through a single-mode microwave radiation-assisted sol-gel method and ambient temperature drying. Because of the ring focusing and power-max technology, the fabrication procedure of carbon aerogels (CAs) are much easier, faster, and cheaper than traditional methods. The RF aerogels were then pyrolysized at 900 °C, and the KOH activation process was used to further dredge micropores in the carbon aerogels. The CAs were characterized by X-ray diffraction (XRD), scanning electron microscopy (SEM), nitrogen adsorption/desorption, and a series of electrochemical tests. The KOH activated carbon aerogels with 3D-nano-network structure exhibited a high specific surface area of 2230 m^2^ g^−1^ with appropriate pore volumes of micro-, meso-, and macropores. The specific capacitance of CAs activated by KOH measured in a two-electrode cell was 170 F g^−1^ at 0.5 A g^−1^ with excellent rate capability and cycle stability in 6 M KOH electrolyte.

## 1. Introduction

Supercapacitors, a unique class of electrical energy storage devices, have attracted tremendous attention in recent years owing to their capability of delivering high power density and remarkable cycling stability [1,2,3,4]. Various materials have been chosen to fabricate electrodes of supercapacitors, such as conducting polymers, metallic oxides, and carbon-based electrode, each of them has both advantages and drawbacks [5,6,7]. Porous carbon nanomaterials, with a hierarchical channel of pores, high surface area, and stable mechanical properties, are attracting increasing interest due to their potential applications in hydrogen storage and electrochemical capacitors [8,9,10,11,12]. 

Carbon aerogels, as one of the 3D porous carbon-based materials, has attracted widespread interest because of their unique three-dimensional nano-network, high conductivity, and the possibility of tailoring their structures to produce final materials that fit the requirements of a specific application [13,14]. Hence, it has been used in various fields, such as catalysis, adsorption, and energy storage [15]. The preparation of CA mainly includes three steps: gelation and curing reactions, drying treatment, and the pyrolysis process [16]. Despite all the advantages, the main drawbacks of this kind of carbonaceous material are that it is time-consuming, and has low specific capacitance and high cost. By traditional methods, it takes at least three days to complete the so-gel process and another several days for the aging procedure. Meanwhile, the supercritical-drying technique required for the drying process is costly and energy consuming [17,18]. Many efforts have been tried to resolve these issues. For instance, Rasines and coworkers have fabricated a highly porous carbon composite composed of N-doped aerogels and carbon black. It displays high surface area and a combined micro/mesopore structure with a desalting capacity of 7.3 mg/g [19]. Chen and coworkers have synthesized a novel porous carbon through facile hydrothermal method to prevent the unexpected re-stacking. The surface area of this 3D graphene/porous CA can achieve 2211 m^2^ g^−1^ and the highest capacity is up to 410 F g^−1^ at a current density of 0.5 A g^−1^ [20]. In order to shorten the synthesis procedure, some researchers have introduced microwave-assisted technology to the heating or drying process to make the fabrication much easier, faster, and cheaper [21,22,23].

In this report, we introduced a novel and simple method to prepare resorcinol-formaldehyde aerogel using a ring focusing single-mode microwave synthesizer. Compared with a multi-mode microwave, the single-mode microwave can make the reaction more accurate, easy to control, and with better repeatability. With the help of ring focusing and power-max technology, the sol-gel process is over 100 to 1000 times faster than traditional heating method. To optimize the specific area and pore size distribution of CAs and make it more suitable as electrode material for supercapacitor, CAs were activated, characterized and used as for supercapacitor application.

## 2. Materials and Methods

### 2.1. Preparation of Carbon Aerogel

An organic precursor was synthesized by polycondensation of resorcinol (R) and formaldehyde (F) using distilled water as solvent and sodium hydroxide as basification catalyst (C). The gel precursor solution was prepared with the molar ration of R/F and R/C fixing at 0.5 and 500 respectively and the mass fraction of the reactants (the total mass of resorcinol and formaldehyde) at 25 wt %. Resorcinol (Aladdin, 99%) and catalyst were first dissolved in deionized water under magnetic stirring. After that, formaldehyde (Aladdin, 37 wt % in water, stabilized by 10.7 wt % methanol) was added to the mixture solution and stirred for an hour. Subsequently, 30 mL of precursor solution was poured into a round-bottom flask and located in the unimode microwave cavity. The sol-gel reaction was carried out under the single-mode microwave radiation at 85 °C and 100 W. The excess solvent in the hydrogel was replaced by acetone for 2 days and then dried at ambient temperature to obtain resorcinol/formaldehyde (RF) organic aerogels. At last, the organic aerogels were pyrolysised in argon flow at 900 °C for 2 h to get the hierarchical porous carbon aerogel (CA). The synthetic process of CA was shown in Scheme 1.

### 2.2. Activation of Carbon Aerogel

The KOH activation reaction was as follows: KOH and CA, with the mass ratio from 1:1 to 5:1 (m_KOH_:m_CA_), were ground adequately in an agate mortar. The resulting mixture was activated in the nitrogen atmosphere at 800 °C for 2 h. After cooling, the products were thoroughly washed by 1M HCl and deionized water to remove all soluble impurities and then dried at 120 °C to obtain activated carbon aerogel. The final products were named as KOH-CA1, KOH-CA2, KOH-CA3, KOH-CA4, and KOH-CA5, respectively.

### 2.3. Materials Characterization

Field-emission scanning electron microscope (FESEM, JSM-6510, JEOL Ltd., Tokyo, Japan) and X-ray powder diffraction (Rigaku D/max 2500/PC, Rigaku Corporation, Tokyo, Japan) were used to identify the morphology and structure of the as-prepared materials. Nitrogen adsorption/desorption isotherms and textural properties were determined at 77 K with a Specific Surface Area and Porosity Analyzer (BELSORP-max, MicrotracBEL, Osaka, Japan).

The electrochemical performance at room temperature of the as-synthesized materials was characterized in 6 M KOH alkaline electrolyte using a CHI660E electrochemical work-station (CHI660E, Huakeputian Technology Co., Ltd., Beijing, China). The electrodes were prepared by pressing a mixture of 85 wt % active material, 10 wt % conductive carbon black and 5 wt % PTFE onto a nickel foam mesh current collector [24,25]. The mass of each electrode was about 5 mg. Cycle voltammetry measurements (CV) and electrochemical impedance spectroscopy (EIS, Huakeputian Technology Co., Ltd., Beijing, China) were performed in three electrodes test system while the galvanostatic charge–discharge measurement (GCD, Huakeputian Technology Co., Ltd., Beijing, China) was carried out using two-electrode cells. The specific capacitance (Csp in F g^−1^) can be calculated from the GCD curve according to the equation [1,26]:(1)$$Csp=\frac{I\Delta t}{\Delta V\times m}$$
where *I* is the constant discharge current (A), Δt is the discharge time (s), ΔV is the working voltage window and m is the mass of the active material of each electrode.

## 3. Results and Discussion

### 3.1. XRD Analysis of CA

Figure 1 shows the XRD patterns of CA prepared with different radiation time. All the samples exhibit a board reflection at 2θ = 23.5° and 43.8°, which are correlated to the (002) and (100) diffraction direction of graphite. It indicates that RF organic aerogel possesses a low degree of graphitization during the carbonization [19,27,28]. The C-C bonds in the carbon aerogel contains both sp^2^ and sp^3^ hybrid, which results in an amorphous nature with local micro crystallites.

### 3.2. SEM Analysis of CA

The SEM micrographs of carbon aerogels (CAs) synthesized by single-mode microwave-assisted polymerization are shown in Figure 2. 

The morphology of CAs displays a 3D-porous network structure formed by spherical nanoparticles. There is a certain gap and channel among the particles which is conducive to the rapid migration of electrolyte ion into the pores of carbon aerogel. This particular structure is suitable to store and release energy and make the CAs a good candidate for electrode materials of supercapacitor.

### 3.3. Electrochemical Properties of CA

To screen the optimal synthesis duration for the CAs, the electrochemical performance of the CAs prepared with different radiation durations was first assessed using a three-electrode test system with Ag/AgCl as reference electrode and Pt as the counter electrode and the results are shown in Figure 3.

In Figure 3a, the as-prepared CAs display a CV curve similar to an ideal rectangular without any redox peak (at a scan rate of 5 mV/s), which illustrates that all the samples demonstrate electric double-layer capacitance and the electrical resistance is low. Figure 3b shows the charge/discharge curves of 30 min-CA, 40 min-CA, 50 min-CA, and 60 min-CA at 0.5 A g^−1^. The GCD tests are carried out using two-electrode cell in 6 M KOH. All the curves display nearly symmetrical isosceles triangles and rapid responses of current–voltage. It indicates that the electrode materials have excellent coulomb efficiency and electrochemical reversibility [1,29]. The 40 min-CA exhibits the smallest ohmic drop with the longest charge and discharge time, which can be concluded that this sample has the lowest equivalent series resistance [20,30] and the highest specific capacitance. (92 F g^−1^ at 0.5 A g^−1^). Figure 3c shows the Nyquist plots of CAs prepared with different radiation times with the frequencies ranging from 0.1 Hz to 100 kHz. All plots compose of a small arc in the high frequency region and a straight line in the low frequency region. The semicircle at high frequency represents the interfacial resistance, charge transfer resistance and the conductivity of electrodes. The bias line at low frequency area reveals the diffusion resistance in the electrolyte [31,32,33,34]. The equivalent series resistance of the 40 min-CA is about 0.47 Ω, which is lower than the 30 min-CA (0.8 Ω), 50 min-CA (0.55 Ω), and 60 min-CA (0.79 Ω), indicating that the sample of 40 min-CA has the lowest interfacial charge-transfer resistance and the highest electrical conductivity. Combined with the results of CV and GCD, we can come to the conclusion that the CA prepared by 40 min microwave radiation exhibits the best electrochemical performance.

### 3.4. XRD Analysis of Activated CA

To further optimize the pore structure and enhance the electrochemical properties of the carbon aerogel, CA was activated by various proportion of solid potassium hydroxide. XRD patterns of CAs activated by different alkali carbon ratio are shown in Figure 4.

With the increasing of the content of KOH, the peak at 2θ = 23.5° remains almost identical while the reflection of 43.8° (2θ) decreases progressively (from m_KOH_:m_CA_ = 4:1). It is known that under high temperature, the carbon framework is etched by the redox reactions of various potassium compounds as chemical activator with carbon. Some graphite grains are consumed in these reactions, which may improve the amorphous structure of the CAs. The hierarchical porous structure of the carbon aerogel is created during this process [35,36].

### 3.5. SEM and BET Analysis of Activated CA

The SEM images of CAs activated by KOH are shown in Figure 5. As expected, after activation by KOH at 800 °C, diameter of the aerogel particles greatly reduces, this leads to the dramatically increase of the specific area. As a result, the contact area of carbon materials with the electrolyte ions enlarged and the active sites increased.

Figure 6 shows the nitrogen adsorption/desorption isotherms and pore size distribution of CA and KOH-CA3.

It can be observed from Figure 6a that the CA displays a type I isotherm and that for the activated carbon aerogel (KOH-CA3) is type IV. The ultra-fast adsorption increases at a low relative pressure of the two samples demonstrates a large quantity of micropores [23,37]. The obvious hysteresis loop ranging from 0.8–1.0 of KOH-CA3 indicates the presence of rich mesoporous [38], which can be further proved by pore size distribution in Figure 6b. After activation, besides the enormous increase in micropores, a moderate amount of meso/macroporous appears in KOH-CA3, which can provide the ion transport channel and ion-buffering reservoir. The specific area and pore characteristics are listed in Table 1. Compared with the 40 min-CA (596 m^2^ g^−1^ and 0.4025 cm^3^ g^−1^), the specific area and pore volume of KOH-CA3 increase to 2230 m^2^ g^−1^ and 1.9770 cm^3^ g^−1^, respectively.

### 3.6. Electrochemical Properties of Activated CA

The electrochemical performance of the activated carbon aerogel was further investigated. Figure 7a–e shows the cyclic voltammetry (CV) curves of activation carbon aerogels at different scanning rates. After activation, even at a scan rate of 100 mv/s, the CV curves of these samples still reserve a near-ideal rectangular shape, indicating a typical double-layer capacitive behavior with the absence of pseudo-capacitance and a good rate capability. Figure 7f displays the galvanostatic charge/discharge curves of 40 min-CA and activated CAs. The linear behavior and symmetrical curve in the potential window (0–1 V) are due to its high electrical conducting of its unique three dimensionally nano-framework. The IR drop can be negligible of all the activated CAs, which demonstrates the high-speed charge propagation of these as-prepared samples. As described in Table 2, the highest capacity (170 F g^−1^ at 0.5 A g^−1^) of the activated CA appears at m_KOH_:m_CA_ = 3:1, which is much higher than the 40min-CA (92 F g^−1^ at 0.5 A g^−1^).

After activation, the equivalent series resistance (ESR) of the KOH-CA3 reduces to only 0.32 Ω (in Figure 8). The cycling stability test of the 40 min-CA and KOH-CA3 are shown in Figure 9.

After 1000 cycles, no noticeable decrease of the specific capacitance can be seen, indicating that activation does not affect the stability of the carbon aerogel and both the two samples have good cycling performance.

## 4. Conclusions

In summary, carbon aerogels with hierarchical porous structure and excellent electrochemical performance have been prepared by carbonization and activation of resorcinol formaldehyde (RF) organic aerogels. The low-cost RF organic aerogels were prepared by using resorcinol and formaldehyde as precursors, sodium hydroxide as a catalyst through a single-mode microwave radiation assisted sol-gel method and ambient temperature drying. The fabrication procedure of carbon gel was much easier, faster, and cheaper due to the ring focusing and power-max technology. The resulting carbon aerogel (KOH-CA3) exhibits an outstanding specific surface area of 2230 m^2^ g^−1^ and possessed a quantity high fraction of micropores and a moderate amount of mesoporous. The specific capacitance tested in two-electrode cell and calculated from galvanostatic charge/discharge curve was 170 F g^−1^ at 0.5 A g^−1^. The as-prepared CAs shows a good rate capability and ideal cycle stability in 6 M KOH electrolyte.

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
