# Peer review of "Preparation of Hierarchical Porous Carbon Aerogels by Microwave Assisted Sol-Gel Process for Supercapacitors"

_polymers, 2019, doi:10.3390/polym11030429_

Round 1

Reviewer 1 Report

This manuscript describes the preparation of porous carbon aerogels by microwave assisted sol-gel process for supercapacitors Although authors conducted several characterizations and electrochemical analyses, some parts of manuscript deserve modifications.

In title, upercapacitors supercapacitors

In experimental section, an organic precursor was prepared by the reaction of resorcinol and formaldehyde, and then the microwave assisted sol-gel reaction was conducted to yield carbon aerogel. The provision of detailed synthetic scheme with chemical structures will be helpful for general readers to understand this work better.  

If possible, elemental analysis (EA) measurement of the final products is also of importance to investigate their chemical compositions (at least for 40min-CA and KOH-CA3).

Author Response

Dear reviewer,

Thank you for your comments and constructive suggestions on our manuscript. We have studied comments carefully and have made correction which we hope meet with approval. The responds to the comments and the main changes in the paper are as followings.

Point 1: In title, upercapacitors → supercapacitors.

Response 1: “upercapacitors” was corrected as “supercapacitors” in title.

Point 2: In experimental section, an organic precursor was prepared by the reaction of resorcinol and formaldehyde, and then the microwave assisted sol-gel reaction was conducted to yield carbon aerogel. The provision of detailed synthetic scheme with chemical structures will be helpful for general readers to understand this work better.

Response 2: The synthetic scheme with chemical structure was added as follow.

Scheme 1. Schematic diagram of synthetic process of CA.

Point 3: If possible, elemental analysis (EA) measurement of the final products is also of importance to investigate their chemical compositions (at least for 40min-CA and KOH-CA3).

Response 3: After carbonization, the final product is an inorganic carbon material composed of almost 100% of C element, so elemental analysis is not necessary in this case. Besides, EA of those products cannot be applied at this moment because the project was finished about 1 year ago and preparing new specimens for EA is time-consuming and difficult.

Reviewer 2 Report

The manuscript “Preparation of hierarchical porous carbon aerogels by microwave assisted sol-gel process for supercapacitors” reported resorcinol formaldehyde carbon aerogels prepared by using resorcinol and formaldehyde as precursors, sodium hydroxide as catalyst through single-mode microwave radiation assisted sol-gel method and ambient temperature drying. The authors claimed that the KOH activated carbon aerogels with 3D-nano-network structure exhibited a high specific surface area and good electrochemical performance. The authors have provided solid data to back up the conclusions in most cases. However, when reading the manuscript some questions arise, therefore some complementary information and revision should be taken into account before being published in “Polymers”.

1.     It should be “supercapacitors” in the title.

2.     In the caption of Fig. 3, (c) and (d) were assigned on the contrary.

3.     In Fig. 7a-e, the same color for each scan rate is suggested for readers to compare.

4.     For carbon materials, the Raman spectroscopy is an effective method to compare the structural difference, which could improve the manuscript further.

Author Response

Dear Reviewer,

 Thank you very much for you comments and constructive suggestions on our manuscript. We have studied comments carefully and have made correction which we hope meet with approval. The responds to the comments and the main changes in the paper are as followings.

Point 1: It should be “supercapacitors” in the title.

Response 1: “upercapacitors” was corrected as “supercapacitors”.

Point 2: In the caption of Fig. 3, (c) and (d) were assigned on the contrary.

Response 2: (c) and (d) were reassigned.

Fig.3. (a) Cyclic voltammograms of the CAs prepared in different radiation time at 5mV s-1 in 6 M KOH solution. (b) Galvanostatic charge–discharge curves using a two-electrode cell at a current density of 0.5 A g-1. (c) Nyquist plot of CAs with the frequency ranging from 0.1 Hz to 100 kHz.(d) Specific capacitance of CAs at 0.5 A g-1.

Point 3: In Fig. 7a-e, the same color for each scan rate is suggested for readers to compare.

Response 3: We are thankful to the reviewer for the suggestion. The CV curves were replotted with the same color for each scan rate in Fig.7a-e.

Point 4: For carbon materials, the Raman spectroscopy is an effective method to compare the structural difference, which could improve the manuscript further.

Response 4: We agree with the reviewer that the Raman spectroscopy is an effective method to compare the structural difference for carbon materials, which could improve the manuscript further. The crystal structure of the materials in this study was performed by XRD. The final products were mainly amorphous carbon aerogels with local microcrystalline structure. The structural differences of the products were not obvious. Furthermore, the morphology and microstructure of the materials were examined by the SEM micrographs and BET analysis. So, it can be concluded that the internal structure of the materials is assessed and the Raman spectroscopy analysis cannot present additional information about structural difference of the carbon aerogels, and accordingly it is not necessary in this case. Besides, Raman spectroscopy analysis of those carbon materials cannot be applied at this moment because the project was finished about 1 year ago and preparing new specimens for Raman spectroscopy analysis is time-consuming and difficult.

Finally, we would like to thank you for their important points and hope the revised version is acceptable for publication.

Yours sincerely,

Name: Dayong Gui

Round 2

Reviewer 1 Report

The manuscript was modified will and now it can be published in Polymers

Author Response

Dear Reviewer,

Thank you for very much for your comments and constructive suggestions on our manuscript.We have studied comments carefully and have made correction which we hope meet with approval. The responds to the comments and the main changes in the paper are as followings.

(1)   In title, upercapacitors → supercapacitors

Response: “upercapacitors” was corrected as “supercapacitors” in title.

(2)   In experimental section, an organic precursor was prepared by the reaction of resorcinol and formaldehyde, and then the microwave assisted sol-gel reaction was conducted to yield carbon aerogel. The provision of detailed synthetic scheme with chemical structures will be helpful for general readers to understand this work better.

Response: The synthetic scheme with chemical structure was added as follow.

Scheme 1. Schematic diagram of synthetic process of CA.

(3)   If possible, elemental analysis (EA) measurement of the final products is also of importance to investigate their chemical compositions (at least for 40min-CA and KOH-CA3).

Response: After carbonization, the final product is an inorganic carbon material composed of almost 100% of C element, so elemental analysis is not necessary in this case. Besides, EA of those products cannot be applied at this moment because the project was finished about 1 year ago and preparing new specimens for EA is time-consuming and difficult.

Yours sincerely,

Name: Dayong Gui
